# Discrimination of False Response from Object Reality in False Belief Test in Preschool Children

**DOI:** 10.3390/jintelligence13100124

**Published:** 2025-09-25

**Authors:** Melis Süngü, Tevfik Alıcı

**Affiliations:** 1Faculty of Economics, Administrative and Social Sciences, Antalya Bilim University, 07190 Antalya, Turkey; 2Faculty of Arts and Science, Bursa Uludağ University, 16059 Bursa, Turkey

**Keywords:** theory of mind, false belief test, object’s location

## Abstract

The first-order false belief (FB) test is frequently employed to assess theory of mind (ToM); however, it faces substantial criticism regarding its inadequacies. Critics argue that the responses remain binary and are influenced by the presence and location of the object. This study aims to address these criticisms by manipulating an object’s location through three alternative FB tasks, thereby enhancing the understanding of children’s reasoning strategies (reality, belief, or perceptual access reasoning) and offering a language skill-independent measure of ToM. This study involved 150 children aged 3–6 years who were administered standard and three alternative FB tasks along with a receptive vocabulary acquisition test. The findings revealed that children predominantly utilized reality reasoning, identifying the object’s location as the correct response. However, in a condition where the object was physically removed, the percentage of correct responses increased significantly, and the use of belief reasoning increased. While age and language skills were found to be directly correlated with FB performance, the object’s interference with belief reasoning in younger children was reduced. In light of these findings, the three alternative tasks are posited to offer a promising, more accurate measure of FB understanding, independent of the object’s presence and language skill.

## 1. Introduction

Theory of mind (ToM) refers to the capacity to interpret and predict the actions of others based on their intentions, beliefs, knowledge, and mental states ([7]). By four years of age, children demonstrate the ability to differentiate their mental processes from those of others ([35]; [36]; [43]). This developmental milestone enables them to adopt others’ perspectives, comprehend what others perceive and interpret, and successfully complete the first-order false belief (FB) test, considered the fundamental measure of ToM abilities ([4]; [45]). The standard FB test is a social cognitive assessment that evaluates children’s ability to differentiate between their own beliefs and those of others, as well as reality, based on unexpected relocations. In a classic paradigm, a child observing a doll places a ball in a basket, and then, in the doll’s absence, the ball is transferred to a box by another agent. Children are then required to predict the location where the doll will search for the ball upon its return to the room ([4]). As a result, the understanding of present reality and false belief diverge; the child must shift from a situation-based comprehension to a representation-based understanding of behavior ([33]; [17]).

Although ToM is a universal social cognitive ability with a consistent developmental trajectory ([2]; [44]), studies have shown a 55% variation in FB test performance ([2]). Empirical evidence suggests that a substantial proportion of ToM variance is shaped by environmental factors ([21]), with the following distribution: 44% non-shared environment, 20% shared environment, and only 15% genetic influence ([22]; [40]). Thus, ToM ability appears to be significantly shaped by experiences unique to the individual, particularly those stemming from non-shared environmental factors. Language skills are also a crucial factor in ToM development. Both FB and language skills develop rapidly in the first five years of life, and it is claimed that there is a bidirectional relationship between them ([42]; [43]). In addition, [26] ([26]) demonstrated that FB test performance is significantly correlated with language skills, regardless of age, suggesting that early language skills can influence later FB performance. Numerous studies have shown that mothers’ language skills and cooperative interactions with siblings have significant links to later ToM abilities in children aged 3–6 years ([6]; [11]; [19]; [25]; [34]). Furthermore, studies on language acquisition in preschoolers have shown that children trained on communication verbs and sentence complements demonstrate improved FB reasoning even without discourse about deception ([20]; [28]). Language appears to provide the representational structures necessary for ToM, as children with language delays are often found to be deficient in this area, even on tasks without a linguistic requirement ([9]). Taken together, these findings indicate that classic FB tests are susceptible to a variety of factors, particularly language skills, and that they are challenging for children, as even minor modifications can significantly impact performance ([5]). Thus, these tests fall short of providing a truly comprehensive measure of ToM.

The primary criticism is that this complex task provides a binary assessment of skills. A child’s response of “basket” is presumed to indicate developed ToM, while a response of “box” is assumed to indicate a lack of it. This limitation is not mitigated by using multiple tasks, as each still presents a binary choice. For instance, children who succeed in one task but fail in another may be doing so due to measurement error, distraction, confusion, or chance, rather than a genuine lack of ToM ([2]). In addition to this binary limitation, research has shown that the specific phrasing of the question can profoundly impact a child’s response. When the question “What will happen next?” was posed instead of the standard “Where does the doll look for the ball?”, the children’s correct response rate increased ([38]). This suggests that constraining the test to a binary response may impede children’s ability to provide accurate responses. Defining mental representation in preschool children at a temporally or situationally stable level is challenging ([41]), and performance varies accordingly.

Research indicates that 2- and 3-year-old children demonstrate the capacity to perceive beliefs and intentions and can engage in perspective-taking ([27]; [29]; [38]). For example, [30] ([30]) observed that 2-year-olds modify their behavior by referencing parents’ mental states regarding the location of a toy. However, despite comprehending that “seeing leads to knowing” ([46]), these children often respond in accordance with reality in the test. This phenomenon can be attributed to several factors. First, the visual information presented in the test (the ball being removed from the basket and placed in the box) may be perceived as more reliable than the doll’s belief. This phenomenon can be attributed to children’s tendency to disregard their practical intelligence when presented with verbal questions, difficulty with inferential questions in passive states, and their propensity to respond to the test solely based on the current reality ([18]; [37]; [47]). However, maintaining focus on the protagonist throughout the narrative and increasing the salience of the protagonist’s mental state facilitates retention of FB and emphasizes the connection between the non-current location and the object. This leads to a significant increase in correct responses ([38]). It is also noteworthy that performance improved when the real object was absent, meaning the salience of real-world content was diminished, and the rate of incorrect responses increased when the salience of the target object was emphasized ([39]; [44]). This suggests that preschoolers’ perspective-taking abilities remain fragile and susceptible to distortion ([10]). This concern has extended to more modern tasks, with a prominent debate showing that the interpretation of children’s behavior in these measures is heavily influenced by methodological details ([1]; [8]). For instance, it has been argued that key differences such as the age of participants, the abstractness of the task’s goal, and whether the goal is explicitly stated or implicitly inferred can lead to contrasting results ([8]). Consequently, a child’s success in such tasks may not be a definitive indicator of true belief reasoning but rather a reflection of their understanding of a specific social rule or a simpler, non-mentalistic strategy ([1]). In the standard FB test, the expected response from the child is to indicate an empty position rather than the current position of the target, which necessitates detachment from the actual presence of the object. However, the salience of the target appears to be a critical factor; when preschoolers justify their judgments, they often refer to the state of the world rather than the character’s mental state.

Furthermore, children may give correct or incorrect responses in tasks without FB understanding. There has been increasing concern that the conventional interpretation of FB test performance may be inadequate and that children are capable of providing correct answers using “perceptual access reasoning” (PAR) without FB reasoning ([13]; [15]). To genuinely comprehend FB, a child must first grasp the concept of mental states and the causal link between ‘seeing’ and ‘knowing’. However, it has been proposed that a standard FB test can be passed without establishing this structure ([14]). Unlike genuine FB reasoning, which relies on an understanding of mental states, PAR is a simpler, rule-based strategy based on perceptual information. For example, a child using PAR might deduce, “The doll did not see the ball being moved, so she doesn’t know it’s in the box,” and arrive at the correct answer without representing the doll’s false belief. Thus, while both strategies can lead to a correct response, they measure fundamentally different cognitive abilities: a true understanding of ToM versus a reliance on perceptual cues. Children can successfully navigate this test by relying on simple perceptual cues rather than by understanding the mental states and thoughts of others ([13]). Children’s knowledge is not denied or underestimated, but the necessity for cognitive change and stability has been emphasized ([44]). [16] ([16]) argued that PAR is a more plausible developmental precursor model because it provides a bridge between an introductory insight into mental states at 2–3 years of age and a fully comprehended understanding of mental states at 7–8 years of age. Consequently, it can be argued that success or failure in the standard FB test does not provide a comprehensive assessment of a child’s cognitive and perceptual abilities.

The present study focuses on the unexpected transfer paradigm not merely due to its widespread use, but because it is uniquely suited for investigating a key confounding variable: the perceptual salience of the object. This paradigm was particularly suitable for our research, as it allowed us to precisely manipulate the key variables of object salience and location. The primary objective of the current study was to address the limitations of the standard FB test by designing and implementing three alternative tasks that would allow us to transcend a mere pass/fail evaluation. This research was designed to clarify the reasoning strategies—belief reasoning (BR), perceptual access reasoning (PAR), and reality reasoning (RR)—employed by children and facilitate a more accurate detection of children with FB understanding. To achieve this, we systematically manipulated the presence and location of the target object across these tasks. In two scenarios, the object (a ball) remained visible, either placed in the ‘correct answer’ location or in a separate location reflecting the standard FB test discrepancy. In a third task, the object was entirely removed from the screen, thereby eliminating its confounding perceptual effect.

Consistent with the existing literature, we first hypothesized that in conditions where the object was visible, children’s performance would be dominated by RR, leading to a high frequency of incorrect responses related to the object’s current location. Also, we hypothesized that by removing the object, the task would provide a purer measure of BR, free from misleading perceptual cues and less confounded by language skills. The absence of a visible object would mitigate the risk of children misinterpreting the FB question as pertaining to the object’s reality, thereby allowing for a more accurate assessment of their true ToM capacity. Furthermore, in line with a developmental perspective, this study examines how children’s performance on these tasks varies with age and language proficiency. We specifically investigate whether the removal of misleading perceptual cues in our no-object condition would alter the established developmental trajectory of FB understanding seen in the standard FB test. Consequently, by creating alternative tasks that account for an object’s location and salience, this study proposes a more reliable approach to FB assessment that provides a comprehensive understanding of the distinct cognitive mechanisms contributing to performance variability.

## 2. Materials and Methods

### 2.1. Participants

A total of 156 parents consented to their child’s participation. However, data from six children were subsequently excluded from the final analysis: two due to a diagnosis of Attention Deficit and Hyperactivity Disorder diagnosis, two who could not be contacted, and two who repeatedly responded with “I don’t know” and refused to engage further. The final sample consisted of 150 preschoolers aged 3–6 years (80 girls and 70 boys) from local preschools in Bursa (Türkiye) province (*M* = 57.67 months, *SD* = 9.61 months, range = 37–78 months) (see Table 1). Due to the presence of only one participant aged 78 months, this individual was excluded from the subsequent age group analyses to ensure more balanced distribution and prevent undue influence from an outlier. The participants were all monolingual native Turkish speakers, predominantly from middle-class families, with 64.7% of mothers and 59.3% of fathers holding a bachelor’s degree, 6.7% of parents being divorced, and 62.7% of children having no siblings. The mean receptive vocabulary score for the sample was 88.84 (*SD* = 20.16), with scores ranging from 34 to 121.

### 2.2. Materials

*Child information form*: A child information form was developed to collect demographic data on both the child and their parents. This form included inquiries about the child’s gender, date of birth, number of siblings, place of residence, age at school commencement, age of initial word utterance and speech onset, and health status, as well as parental education level. This form was provided to parents concurrently with the informed consent form.

*Receptive vocabulary acquisition*: The receptive vocabulary acquisition was assessed using the Turkish version of the Peabody Picture Vocabulary Test (PPVT), which is a standardized measure of vocabulary development in children aged 3–17 years ([12]; [23]; [31]). The Cronbach’s alpha coefficients for the Turkish version ranged from 0.91 to 0.95 ([32]). For each trial, the examiner articulated a word, and the child was required to indicate the corresponding image. No time constraints were imposed on this study. The initial card was selected based on the participant’s age, and the test proceeded until the child made six errors within eight questions. The participants’ raw scores were calculated by subtracting the number of words to which they responded incorrectly. Utilizing the Receptive Language Age Calculation Table, this raw score was converted into the child’s receptive-language age based on their place of residence (village, city, shanty).

*False belief tasks*: Standard FB test and alternative test (3 tasks) were variations of the well-established expected and unexpected transfer paradigms. The Standard FB test served as the baseline. In this classic paradigm, a doll is placed in a scene with a ball (the object), a box, and a basket ([3]; [24]). After placing the ball in the basket, the doll exits the room. Subsequently, the ball is moved to the box in the doll’s absence. Children observe all events and are presented with FB questions upon the doll’s return (“Where does the doll look for the ball?”). Subsequently, the children are asked the memory question (“Where did the doll put the ball initially?” “Where was the ball placed thereafter?”), the reality question (“Where is the ball currently located?”), the information question (“Is the doll aware of the ball’s location?”), and the look-first question (“Upon the doll’s return to the room, where does she initially search for the ball?”).

The alternative version of the FB test systematically manipulated key perceptual features to elucidate children’s reasoning strategies. In the three alternative tasks, the scenario involved a doll, a ball, a box, and a basket. Initially, the doll first placed the ball in the basket but then explicitly requested that the experimenter move the ball to the box in its absence. The experimenter then altered the ball’s location across three distinct conditions in the absence of the doll: (1) Box Condition (Reality and Belief Converge): The ball was moved to the box as requested. Both BR and RR would lead to the correct response of “box.” A response to the basket was coded as PAR. (2) Basket Condition (Reality and Belief Diverge): The ball remained in the basket, contrary to the doll’s request. In this condition, the doll’s false belief is based on an unfulfilled expectation rather than a lack of information. A BR response would indicate the “box” (where the doll believes the ball to be), while an RR response would indicate the “basket” (the ball’s actual location). (3) No-Object Condition (Perceptual Cues Eliminated): The ball was removed from the basket and entirely taken out of the scene. A RR response would typically be “outside the scene” or “gone.” whereas a BR response would indicate the “box.” PAR users were considered to split their responses between ‘box’ and ‘basket’ as both were empty and equally incorrect from a purely physical perspective, reflecting a lack of specific belief-based reasoning towards an empty location.

Following each task, the same battery of questions as in the standard FB test was administered. As the order of the questions was not significant ([14]), the FB question (‘Where does the doll look for the ball?’) was always presented first after familiarization with the task objects.

### 2.3. Procedures

To ensure consistency and control, all tasks were pre-recorded with a single experimenter using toys on a white table against a white background. The narration was deliberately repetitive and slow to ensure its suitability for all age groups. To mitigate potential bias resulting from the consistent placement of the box and basket, each video was reversed by mirroring, yielding a total of eight videos. Each task concluded when the doll reappeared in the video, after which the experimenter proceeded to ask the questions. Before the experiment commenced, participants were first familiarized with the toys (doll, ball, basket, and box). In a single, 30 min session, the participants completed all four FB tasks and the PPVT at their respective schools. The order of the four FB tasks was randomized for each participant, and the PPVT was administered either at the beginning or the end of the session. All participants were assessed by the same researcher.

### 2.4. Coding and Data Analysis

Responses to the FB question were coded based on the location indicated by the child, allowing for the classification of underlying reasoning strategies: Correct Response (1): Reflecting BR (e.g., “basket” in the standard FB test, “box” in the alternative tasks). Object’s Current/Final Location (2): Reflecting RR (“box” in the standard FB test and box condition, “basket” in the basket condition, “outside the scene” or “gone” in the no-object condition). Object’s Initial Location (3): Reflecting the first placement of the object (“basket” in box and no-object condition). Other Responses (4): Any other non-categorized response.

Given that diverse reasoning strategies (BR, PAR, and RR) could consistently produce accurate responses in the various tasks, a direct comparison based solely on ‘correct answer’ proportions proved insufficient for elucidating children’s fundamental reasoning strategies. Following the categorization of responses, task-specific estimated proportions for each reasoning strategy (BR, RR, PAR) were calculated by determining the frequency with which responses indicative of a particular strategy occurred within each FB task. Specifically, the overall average proportion for RR was calculated as the mean of its task-specific estimates across the conditions where RR-indicative responses were present (standard FB test, basket condition, and no-object condition). Similarly, the overall average proportion for PAR was calculated as the mean of its task-specific estimates from the conditions in which PAR-indicative responses were identifiable (box condition and no-object condition). The remaining proportion, not accounted for by RR or PAR in the overall average, was attributed to the BR. The derived overall average proportions for RR, PAR, and BR were then used as a baseline for subsequent comparisons to understand the prevalence and shifts in children’s reasoning strategies across different task conditions. In addition to this descriptive analysis, McNemar analysis was employed to compare the performance between the different experimental conditions (standard FB, basket, box, and no-object). Subsequently, a multinomial logistic regression analysis was conducted to assess the effects of age and language variables, both individually and in combination, on each FB tasks. Finally, analysis of variance (ANOVA) was used to identify the age group at which FB performance began to improve.

## 3. Results

The task-specific estimated proportions for each reasoning strategy (BR, RR, PAR) were calculated as follows. Consistent with previous findings, RR was a prominent strategy across conditions. In the standard FB test, children predominantly chose the box (0.68 of the time), reflecting the actual object location. Similarly, children identified the object’s current location in the basket in the basket condition (0.50 of the time), and selected ‘outside the scene’ in the no-object condition (0.42 of the time), further indicating RR. PAR also featured in children’s response patterns. In the box condition, PAR users selected the basket at an estimated proportion of 0.16. For the no-object condition, the choice of the basket (0.14 of the time) reflected a PAR estimate of 0.28, considering children’s split responses between empty locations. Based on these task-specific estimates, the overall average proportion for each reasoning strategy across all tasks was determined. The overall average proportion for RR was 0.53. For PAR, it was 0.22. The remaining proportion of 0.25 was attributed to BR. These overall average proportions for RR (0.53), PAR (0.22), and BR (0.25) served as the baseline for subsequent analyses comparing choice rates across task conditions. Detailed responses across the FB tasks are presented in Table 2.

### 3.1. The Differences in Responses to Four FB Tasks

As predicted, only BR and PAR users were expected to provide correct answers in the standard FB and basket conditions. The predicted correct response rate for both conditions was 0.47 (0.25 + 0.22). However, when both conditions are compared in terms of correct and object’s location responses, it is seen that these children do not consistently give the same answers. In the standard FB test, the observed correct response rate was only 0.32, which was lower than the predicted rate. In contrast, in the basket condition, the observed correct response rate was nearly 0.50, which aligned closely with the predicted rate. This finding suggests that despite the similarity in task structure, the proportion of children who answered the standard FB test correctly was significantly lower than those who responded correctly in the basket condition. A McNemar analysis confirmed this discrepancy. When comparing correct and object’s location responses between the standard FB test and the basket condition, the basket condition yielded a significantly higher number of correct answers than the standard FB test, *χ*^2^ (1) = 0.314, *p* < .001 (see Table 3).

The predicted rate for the no-object condition was 0.36 (0.25 + 0.11) as only BRs and half of the PARs were expected to give the correct answer. When the standard FB test and no-object conditions were compared in terms of correct and object’s location responses, the estimated rate of 0.47 for the standard FB test was observed to be 0.25. In the no-object condition, the observed rate was 0.44, which was above the estimate. A McNemar analysis also confirmed that the correct answers of children increased in the no-object condition. In the absence of the object, the percentage of correct answers increased significantly compared to the standard FB test, *χ*^2^ (1) = 0.319, *p* < .001 (see Table 4). It was observed that only 37 of these children gave correct answers in the standard FB test, while 27 more were able to reach the correct answer when the object was removed.

As predicted, there was no significant difference in response patterns when comparing the box condition and the standard FB test, *χ*^2^ (1) = 0.029, *p* = .864. In the box condition the predicted rate was 0.78 (0.25 + 0.53) since both BR and RR would lead to the correct answer. In this case, because the box condition included both answers, a comparison with the standard FB test could not be made, and the observed rate for the box was found to be 0.84, which was as high as expected.

**Table 2 jintelligence-13-00124-t002:** Number of responses of age groups to each false belief task.

Age; Month/N	Standard FB Test			Box Condition			Basket Condition		No-Object Condition			
	Correct (Basket)	False (Object’s Location-Box)	False	Correct (Box)	First Location (Basket)	False	Correct (Box)	Object’s Location (Basket)	Correct (Box)	Object’s Location (Outside of the Scene)	First Location (Basket)	False
**3** **–** **3; 5** **N = 11**	1 (9.1%)	10 (90.9%)	0	9 (81.8%)	1 (9.1%)	1 (9.1%)	4 (36.4%)	7 (63.6%)	4 (36.4%)	6 (54.5%)	1 (9.1%)	0
**3; 6** **–** **3; 11** **N = 11**	1 (9.1%)	10 (90.9%)	0	10 (90.9%)	1 (9.1%)	0	4 (36.4%)	7 (63.6%)	0	8 (72.7%)	3 (27.3%)	0
**4** **–** **4; 5** **N = 24**	5 (20.8%)	19 (79.2%)	0	21 (87.5%)	3 (12.5%)	0	7 (29.2%)	17 (70.8%)	8 (33.3%)	14 (58.3%)	1 (4.2%)	1 (4.2%)
**4; 6** **–** **4; 11** **N = 38**	9 (23.2%)	29 (76.3%)	0	30 (78.9%)	8 (21.1%)	0	14 (36.8%)	24 (63.2%)	10 (26.3%)	21 (55.3%)	7 (18.4%)	0
**5** **–** **5; 5** **N = 29**	14 (48.3%)	14 (48.3%)	1 (3.4%)	22 (75.9%)	6 (20.7%)	1 (3.4%)	17 (58.6%)	12 (41.4%)	16 (55.2%)	8 (27.6%)	5 (17.2%)	0
**5; 6** **–** **5; 11** **N = 28**	14 (50%)	14 (50%)	0	25 (89.3%)	3 (10.7%)	0	22 (78.6%)	6 (21.4%)	21 (75%)	4 (14.3%)	3 (10.7%)	0
**6** **–** **6; 5** **N = 8**	3 (37.5%)	5 (62.5%)	0	7 (87.5%)	1 (12.5%)	0	7 (87.5%)	1 (12.5%)	6 (75%)	2 (25%)	0	0

**Table 3 jintelligence-13-00124-t003:** Results of the McNemar analysis of the standard FB test and basket condition.

		Basket Condition		Total	*χ* ^2^	*p*
		Correct Response	Object’s Location			
**Standard FB test**	Correct response	36	12	48	0.31	.000 ***
	Object’s location	39	61	100		
**Total**		75	73	148		

*** *p* < .001.

**Table 4 jintelligence-13-00124-t004:** Results of McNemar’s analysis of the standard FB test and no-object condition.

		No Object		Total	*χ* ^2^	*p*
		Correct Response	Object’s Location			
**Standard FB test**	Correct response	31	6	37	0.31	.000 ***
	Object’s location	33	58	91		
**Total**		64	64	128		

*** *p* < .001.

### 3.2. The Effect of Age and Receptive Vocabulary Acquisition on FB Performance

A multinomial logistic regression analysis was employed to investigate the impact of age and receptive vocabulary acquisition on FB performance. Prior to conducting the primary analyses, we examined the relationship between age and receptive vocabulary acquisition to address potential issues of multicollinearity. In the analysis of the Pearson correlation between variables, a strong correlation (*r* = 0.629) was identified between age and receptive vocabulary acquisition, with the Variance Inflation Factor (VIF) calculated as 1.654. These data indicate multicollinearity between these two variables. Consequently, to accurately assess the unique and combined effects of age and receptive vocabulary acquisition, we adopted a sequential approach in our regression models, first entering the variables individually and then simultaneously. In the model, the correct answers were designated as the reference category.

In the standard FB test, both age (*χ*^2^ (2) = 12.674, *p* = .002) and receptive vocabulary acquisition (*χ*^2^ (2) = 7.350, *p* = .025) were significant independent variables. As shown in Table 5, for each additional month of age, the ratio of object location responses decreased by 7% relative to the correct response; *Exp* (*B*) = 0.93, *p* = .001. Furthermore, for each increment in receptive vocabulary acquisition, the ratio of object location responses decreased by 3%; *Exp* (*B*) = 0.97, *p* = .013. This finding confirms a core principle of FB development; as children age and their language skills mature, they are increasingly able to disengage from the physical reality of the object and correctly reason about the doll’s false belief. When both variables were included in the model, their combined effect significantly predicted FB performance; *χ*^2^ (4) = 13.06, *p* = .011. This result indicated that these variables collectively account for the differences in response observed in the standard FB test performance. In this combined model, the unique influence of age on FB performance was found to be marginally significant (*χ*^2^ (2) = 5.711, *p* = .058), suggesting that, when controlling for receptive vocabulary acquisition, age may have a limited unique contribution to the probability of giving different responses. Each monthly increase in age was associated with a 6% decrease in the object location response compared to the correct response; *Exp* (*B*) = 0.94, *p* = .021. Conversely, receptive vocabulary acquisition did not have a significant unique effect when age was controlled for (*χ*^2^ (2) = 0.388, *p* = .824), indicating that when age was controlled for, language skill did not exert a significant unique effect on the likelihood of classification into different response categories of the standard FB test.

In the box condition, we found no significant effect on the correct answer for age (*χ*^2^ (2) = 0.057, *p* = .972) and receptive vocabulary acquisition (*χ*^2^ (2) = 4.260, *p* = .119) on the correct response rate. Similarly, when the variables were analyzed collectively, the model did not achieve statistical significance; *χ*^2^ (4) = 6.806, *p* = .147.

In the basket condition, both age (*χ*^2^ (2) = 24.993, *p* < .001) and receptive vocabulary acquisition (*χ*^2^ (2) = 11.675, *p* = .003) were identified as significant individual predictors of performance. With each additional month of age, the likelihood of providing the object’s location as a response decreased by 8% compared to providing the correct answer; *Exp* (*B*) = 0.916, *p* < .001. Furthermore, for every unit increase in receptive vocabulary acquisition, the probability of giving the object’s location as the response diminishes by 3%; *Exp* (*B*) = 0.972, *p* = .002. The robust effect of age and language in this condition highlights that, even when a belief-based response contradicts the object’s physical location, children with more developed cognitive and linguistic skills are better able to ignore the perceptual reality and correctly infer the doll’s belief. When both variables were included in the model, they significantly predicted performance; *χ*^2^ (4) = 33.570, *p* < .001. When controlling for receptive vocabulary acquisition, age had a unique contribution to the probability of responding differently in FB performance; *χ*^2^ (2) = 21.895, *p* < .001. Notably, each additional month of age was associated with an 8% reduction in the likelihood of responding to the object’s location as opposed to the correct answer; *Exp* (*B*) = 0.92, *p* = .002 (see Table 6). However, the observed statistical significance for receptive vocabulary acquisition (*χ*^2^ (2) = 8.577, *p* = .014) was not considered reliable, as the 95% confidence interval of the associated odds ratio included 1 (OR = 0.996, 95% CI [0.974–1.020], *p* = .761).

In the no-object condition, age was independently analyzed and identified as a significant variable, *χ*^2^ (3) = 22.344, *p* < .001. With each additional month, there was a 9% decrease in the probability of giving the object’s location instead of the correct answer (*Exp* (*B*) = 0.91, *p* < .001) and a 7% decrease in the probability of giving the object’s first location instead of the correct answer; *Exp* (*B*) = 0.93, *p* = .021. Receptive vocabulary acquisition was also found to be a significant variable, *χ*^2^ (3) = 21.529, *p* < .001. Each additional month was associated with a 4% decrease in the likelihood of responding with the object’s location rather than the correct answer (*Exp* (*B*) = 0.96, *p* < .001), and a marginally decrease of 3% in the likelihood of responding was found in the object’s initial location (*Exp* (*B*) = 0.97, *p* = .054). In this condition also the strong predictive power of age and language suggests that children with more mature skills are better equipped to engage in abstract mental state reasoning. When these variables were examined concurrently, the model was also determined to be significant, *χ*^2^ (6) = 29.170, *p* < .001. However, within this combined model, receptive vocabulary acquisition did not demonstrate a significant unique contribution when controlling for age; *χ*^2^ (3) = 6.826, *p* = .078. Conversely, age retained a marginally significant unique contribution to performance even when controlling for receptive vocabulary (*χ*^2^ (3) = 7.641, *p* = .054), Specifically, each additional month of age was associated with a 7% reduction in the likelihood of responding to the object’s location as opposed to the correct answer; *Exp* (*B*) = 0.93, *p* = .009 (see Table 7).

To provide a crucial developmental context for our findings, and to identify the specific age range at which children’s FB performance begins to improve significantly, we conducted a one-way ANOVA. Participants were stratified into six age groups, and responses to the standard and alternative FB tests, along with receptive vocabulary acquisition, were analyzed by age. A marginally significant difference was observed between the age groups for the standard FB test, *F* (6,142) = 1.955, *p* = .076. From the age of five, the number of correct responses began to increase (*M* = 1.59, *SD* = 0.682). Conversely, no significant age-related differences were found in the box condition, *F* (6,142) = 0.552, *p* = .768. In the basket condition, significant age-related differences were detected, *F* (6,142) = 2.900, *p* = .011. Post-hoc analysis revealed that the 5; 6–5; 11 (*M* = 1.21, *SD* = 0.42) age groups and older participants exhibited significantly higher accuracy in their responses. Similarly, significant differences were also observed in the no-object condition, *F* (6,142) = 2.900, *p* = .001. Although not statistically significant, a clear increase in correct responses was observed in the 3–4-year-old age groups. The number of correct responses increased from the age of 5; 0–5; 6 (*M* = 1.62, *SD* = 0.77), and the post-hoc analysis (Bonferroni correction) indicated that the response accuracy of the 5; 6–5; 11 (*M* = 1.36, *SD* = 0.68) age group was significantly higher than that of the lower age groups.

Lastly, the one-way ANOVA results confirmed that receptive vocabulary acquisition was significantly influenced by age, *F* (6,142) = 17.228, *p* < .001. Receptive vocabulary acquisition increased with age, particularly after the age of 4; 6 (*M* = 90.18, *SD* = 16.86).

## 4. Discussion

The objective of this study was to elucidate the nature of FB skills based on object’s presence and location, and to understand the strategies employed in these tasks. Consequently, we endeavored to devise a more viable FB test in which the roles of salience and object location could be more distinctly observed and demonstrated in preschoolers.

We also hypothesized that manipulating the object’s presence and location would significantly alter children’s performance on FB tasks. Consistent with this expectation, comparative analysis revealed significant disparities between the standard FB test and alternative test. A significant finding was that a subset of children who failed the standard FB test demonstrated correct responses on the alternative tasks. For instance, the basket condition—where the doll’s expected action (the ball being moved) did not occur—elicited a higher rate of correct responses compared to the standard FB test, where the false belief was based on unawareness of an unanticipated location change. Our results underscore the nuanced and context-dependent nature of children’s reasoning, indicating that their ability to engage in belief-based reasoning is not a rigid skill but is profoundly affected by minor changes in the situational context.

A comparison between the standard FB test and the no-object condition also corroborated our primary hypothesis. The no-object condition was designed to disrupt the binary response paradigm by preventing children from relying the object’s current position. Our results showed that children who failed the standard FB test were often able to solve the task in the no-object condition, and a portion of those who typically used RR and PAR began providing correct responses. This results strongly suggest that the object’s physical presence serves as an impediment to BR. These findings are consistent with those of studies examining the reliability of visual information in children and those revealing that performance improved when the salience of the object decreased or when the real object was absent ([18]; [37]; [39]; [44]; [47]). Our findings support the idea that children do not rely on a single reasoning strategy but rather exhibit flexibility depending on the situational context. This flexibility highlights a key limitation of the standard FB test: it fails to distinguish between a genuine lack of ToM and the misleading influence of a salient visual cue. Therefore, the no-object condition provides a purer and more accessible measure of FB understanding, allowing for a more accurate assessment of children’s true ToM capacity.

The present study contributes to this debate by providing evidence that directly addresses the methodological challenges inherent in FB assessment ([1]; [5]; [8]; [13]). Our findings support the argument that a shift in the field is necessary, as simply developing new test methods is insufficient to address the core theoretical problem. The true issue lies in the theoretical framework used to interpret findings. Our results are consistent with an action-based perspective on social cognitive development, which suggests children’s progress is a matter of learning how to participate in different social situations, rather than solely focusing on their ability to attribute mental states. This theoretical shift—from understanding what a child believes to understanding how they represent the interactive affordances of objects and people—is crucial for moving beyond the limitations of existing paradigms and gaining a more accurate understanding of children’s ToM abilities.

Consistent with our hypothesis, we found that performance on FB tasks significantly increased with age and language proficiency. Independent analyses of FB tasks, considering age and receptive vocabulary acquisition, identified both as significant determinants of FB performance. In consideration of all FB tasks, the probability of providing the correct response, as opposed to the incorrect response, consistently increased with age. Similarly, when analyzed independently, the language proficiency also enhanced the probability of a correct response. When both age and receptive vocabulary acquisition were simultaneously included in the model, these factors collectively explained a significant portion of the variance in responses to FB tasks. However, the strong correlation and multicollinearity between age and receptive vocabulary acquisition complicated the interpretation of their unique contributions in our models. While age continued to account for a substantial portion of the variance, its effect seemed to overshadow the unique contribution of language skills. These findings provide strong support for our hypothesis, demonstrating that more mature cognitive and linguistic skills are critical for children’s success in navigating the complexities of FB tasks.

Previous studies consistently reported a significant improvement in standard FB test performance around four years of age ([4]; [36]; [45]), findings echoed in studies with Turkish children ([24]). In our study, however, while we observed an age-related increase in response accuracy, a notable difference emerged closer to five years of age. Additionally, the response rate to the standard FB test was lower than typically documented in the literature. This disparity might be attributed to several factors, including generational and individual differences among children, as well as potential limitations arising from dividing the sample into distinct age groups, which could have resulted in a limited sample size per group. Furthermore, the combined administration of multiple FB tasks might have increased the overall cognitive demands on participants, potentially contributing to a reduction in observed correct responses across tasks. Nevertheless, our developmental findings offer more than a mere confirmation of established trends. The observed age-related performance differences suggest that children’s FB capacity only becomes apparent at a later age in the presence of misleading cues. In contrast, the no-object condition, by removing this impediment, reveals a potentially more authentic developmental trajectory. This suggests that the no-object task may effectively mitigate some of the inherent difficulties associated with the standard FB test for children, offering a potentially more accessible measure of belief understanding. Therefore, these findings reframe these developmental changes as a process of overcoming methodological biases, and underscore that the criticisms of the standard FB test are indeed justified. The presence of the object appears to constitute an impediment, and when this impediment is eliminated, children are able to successfully complete the task. This supports the idea that a child’s failure on a single task may not indicate a genuine lack of ToM, but rather a susceptibility to the misleading influence of salient visual information.

As anticipated, receptive vocabulary acquisition demonstrated a significant improvement with age, with marked differences appearing from 4;6 years. Interestingly, however, significant differences in FB understanding were observed approximately 6–12 months after this age threshold for language development. This temporal lag between advancements in receptive vocabulary and the emergence of robust FB understanding aligns with the existing literature suggesting that language skills may serve as a crucial precursor to comprehensive FB comprehension ([26]), and it further supports the notion that the standard FB task is a linguistically and cognitively demanding test that requires a more mature and integrated set of skills to overcome its inherent biases.

It is also imperative to acknowledge several limitations that warrant consideration for future research. A key limitation of the present study is our exclusive focus on the unexpected transfer paradigm. While this design was uniquely suited for our research objective—to investigate the confounding effect of an object’s perceptual salience—it is important to note that our findings may not generalize to other classic FB tasks, such as the unexpected contents or unexpected identity tasks. Future research should therefore aim to replicate our findings using these tasks to determine whether the confounding effect of perceptual salience and the observed developmental trajectory are consistent across different FB paradigms. The duration of the task and the controlled experimental setting may have influenced children’s attention and the ecological validity of the results. Future studies should consider adapting tasks into more engaging formats, such as animations. Additionally, the scope of the assessment was confined to first-order FB understanding and receptive vocabulary; therefore, incorporating broader measures, including executive functioning and higher-order ToM components, would yield more comprehensive insights. Lastly, although the overall sample size was substantial, future research would benefit from larger and more diverse samples within specific age groups and varying socioeconomic and cultural backgrounds to enhance the generalizability and statistical power of developmental analyses.

Based on our findings, the alternative method developed in this study provides a more effective and precise way to assess FB understanding in preschool children. By differentiating the reasoning strategies utilized by children more effectively, the evaluation of children with a developed ToM who employ BR can be conducted with greater precision. This study offers a significant contribution to the literature by demonstrating that a child’s failure on a single task may not indicate a lack of underlying ToM skills, but rather a susceptibility to visual information. These findings have important practical implications: the developed alternative test, by separating itself from the ambiguities of the standard FB test, can identify potential delays in children’s ToM skills more clearly and at an earlier stage. Consequently, educators and developmental psychologists should consider using multiple assessment tasks to avoid misinterpreting a child’s cognitive abilities, thereby allowing for the initiation of more effective and needs-based educational or intervention strategies, such as social skills and communication therapy programs, at an earlier age.

## Figures and Tables

**Table 1 jintelligence-13-00124-t001:** Descriptive information about the participants.

Age Group (Year; Month)	N			Age (Month)	Receptive Vocabulary Acquisition	
	Girl	Boy	Total	Mean (*SD*)	Min–Max	Mean (*SD*)
**3–3; 5**	8	3	11	39.36 (1.50)	34–97	65 (22.93)
**3; 6–3; 11**	7	4	11	44.45 (1.91)	38–94	71.55 (15.38)
**4–4; 5**	13	11	24	50.29 (1.80)	36–97	73.63 (15.84)
**4; 6–4; 11**	18	20	38	56.47 (2.07)	36–113	90.18 (16.86)
**5–5; 5**	17	12	29	62.07 (1.87)	34–113	95.28 (14.26)
**5; 6–5; 11**	14	14	28	68 (2.95)	85–121	104.71 (10.51)
**6–6; 5**	3	5	8	74.25 (1.28)	65–119	104.88 (18.53)

**Table 5 jintelligence-13-00124-t005:** Results of multinomial regression analysis of the standard FB test.

Variables	Standard FB Test Reference Category: Correct Answer	Age Without Receptive Vocabulary Acquisition				Receptive Vocabulary Acquisition Without Age				Both Age and Receptive Vocabulary Acquisition			
		B	SE	*p*	OR	B	SE	*p*	OR	B	SE	*p*	OR
**—**	Object’s location												
**Age**		−0.069	0.021	.001 ***	0.933	—				−0.060	0.026	.021 *	0.942
**Receptive vocabulary acquisition**		—				−0.026	0.010	.013 *	0.975	−0.007	0.012	.993	0.969
**—**	Wrong												
**Age**		0.020	0.120	.869	0.020	—				0.004	0.152	.980	1.004
**Receptive vocabulary acquisition**		—				0.018	0.071	.804	1.018	0.015	0.083	.861	1.015
**—**	Model General Fit												
***χ*^2^ (df)**		*χ*^2^ (2) = 12.674				*χ*^2^ (2) = 7.350				*χ*^2^ (4) = 13.061			
** *p* ** **-value**		.002 **				.025 *				.011 *			

* *p* < .05, ** *p* < .01, *** *p* < .001.

**Table 6 jintelligence-13-00124-t006:** Results of multinomial regression analysis of basket condition.

Variables	Basket Condition Reference Category: Correct Answer	Age Without Receptive Vocabulary Acquisition				Receptive Vocabulary Acquisition Without Age				Both Age and Receptive Vocabulary Acquisition			
		B	SE	*p*	OR	B	SE	*p*	OR	B	SE	*p*	OR
**—**	Object’s location												
**Age**		−0.087	0.021	.000 ***	0.916	—				−0.082	0.026	.002 **	0.921
**Receptive vocabulary acquisition**		—				−0.028	0.009	.002 **	0.972	−0.004	0.012	.761	0.996
**—**	Wrong												
**Age**		0.0295	0.248	.243	0.343	—				13.19	0.660	.984	0.535
**Receptive vocabulary acquisition**		—				−0.062	0.042	.142	0.940	−3.40	0.625	.996	0.033
**—**	Model General Fit												
***χ*^2^ (df)**		*χ*^2^ (2) = 24.993				*χ*^2^ (2) = 11.675				*χ*^2^ (4) = 33.570			
***p*-value**		.000 ***				.003 **				.000 ***			

** *p* < .01, *** *p* < .001.

**Table 7 jintelligence-13-00124-t007:** Results of multinomial regression analysis of no-object condition.

Variables	No-Object Condition Reference Category: Correct Answer	Age Without Receptive Vocabulary Acquisition				Receptive Vocabulary Acquisition Without Age				Both Age and Receptive Vocabulary Acquisition			
		B	SE	*p*	OR	B	SE	*p*	OR	B	SE	*p*	OR
**—**	Object’s location												
**Age**		−0.092	0.022	.000 ***	0.912	—				−0.070	0.027	.009 **	0.932
**Receptive vocabulary acquisition**		—				−0.039	0.011	.000 ***	0.962	−0.018	0.013	.174	0.982
**—**	Object’s first location												
**Age**		−0.068	0.029	.021 *	0.935	—				−0.056	0.036	.125	0.946
**Receptive vocabulary acquisition**		—				−0.028	0.014	.054	0.973	−0.010	0.018	.574	0.990
**—**	Wrong												
**Age**		−0.172	0.123	.163	0.842	—				−0.047	0.123	.703	0.954
**Receptive vocabulary acquisition**		—				−0.273	0.358	.445	0.761	−0.257	0.356	.471	0.774
**—**	Model General Fit												
***χ*^2^ (df)**		*χ*^2^ (3) = 22.344				*χ*^2^ (3) = 21.529				*χ*^2^ (6) = 29.170			
***p*-value**		.000 ***				.000 ***				.000 ***			

* *p* < .05, ** *p* < .01, *** *p* < .001.

## Data Availability

The datasets used and/or analyzed during the current study are available from the corresponding author on reasonable request.

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
