# Peer review of "Discrimination of False Response from Object Reality in False Belief Test in Preschool Children"

_jintelligence, 2025, doi:10.3390/jintelligence13100124_

Round 1

Reviewer 1 Report

Comments and Suggestions for Authors

The authors have responded well to my comments and I am now happy to recommend publication.

Author Response

Dear Reviewer,

We are writing to express our sincere gratitude for the reviewer's valuable feedback and positive evaluation of our manuscript.

We are delighted to accept the recommendation for publication. The reviewer's comments were constructive and instrumental in significantly improving the quality of our work.

Thank you for your guidance and support throughout the review process.

Sincerely,

Melis Süngü

Reviewer 2 Report

Comments and Suggestions for Authors

Thank you for the opportunity to read about this study.  It was well written but there are ways to improve. 

The beginning could use some structure.  We go from a long section on an introduction right into methods. There are no research questions or rationale purpose of the study stated. 

The multinomial logisitc regression was done well.  Spend more time unpacking what those results mean. 

I am unsure of the reason for one small paragraph on correlation other than to show multicollinarity and no description as to what those implications are.  The ANOVA analysis seemed to be unnecessary unless the goal is to try and see at what age children can be correct vs incorrect, but again that wasnt described. 

I suggest clear research questions, rationale behind each analysis pointing back to research questions, clear description of the answer to each RQ, and then unpacking in the results.  This would provide a much clearer understanding of the purpose of the study and the results. 

Comments on the Quality of English Language

Some of the phrases felt a little odd or peculiar.  For example, we do not refer to an analysis by the name in the possive, (i.e. McNemar's analysis). 

Also unsure of the use of the word Kindergarten.  Ages 3-6 pre-schoolers cannot come from Kindergarten.  Kindergarten is a class.  

I would rely more on a Native English speaker who understands content and context of the paper throughout rather than any machine translations, which it seems like machines were used.  

Author Response

Dear Reviewer,

Thank you for your valuable feedback and insightful comments on our manuscript. We appreciate the time and effort you dedicated to reviewing our work. Your suggestions have been instrumental in improving the clarity and overall structure of our paper.

We have carefully considered all of your points and have revised the manuscript accordingly. Below is a detailed, point-by-point response outlining the changes we have made.

Comment 1: The beginning could use some structure. We go from a long section on an introduction right into methods. There are no research questions or rationale purpose of the study stated.

Response 1: Thank you for this valuable and constructive feedback. We agree with the reviewer's assessment that the original Introduction lacked a clear and structured statement of the study's purpose and hypotheses. We have now extensively revised the final paragraph of the Introduction to address this concern directly. We have now presented the research hypotheses in a structured, making our research questions and predictions clear and easy to follow.

Comment 2: The multinomial logistic regression was done well. Spend more time unpacking what those results mean.

Response 2: We fully agree with your observation that our results required a more in-depth discussion. As you noted, simply presenting the statistical outcomes is insufficient. We have therefore revised the Results sections to provide a more thorough interpretation of our findings, particularly those from the multinomial logistic regression.

Comment 3: I am unsure of the reason for one small paragraph on correlation other than to show multicollinearity and no description as to what those implications are. 

Response 3: We appreciate your point regarding the ambiguity of the correlation analysis's purpose. We have now revised this section to be more explicit. The purpose of the correlation analysis is now stated to have been to identify multicollinearity between age and receptive vocabulary, which in turn justifies our approach of analyzing the variables both individually and in combination within the regression models. This clarification provides a clearer rationale for this methodological step.

Comment 4: The ANOVA analysis seemed to be unnecessary unless the goal is to try and see at what age children can be correct vs incorrect, but again that wasn’t described.

Response 4:  We thank you for raising this important point about the purpose of the ANOVA analysis. We agree that its inclusion was not well-justified in the original manuscript. We have now clarified that the ANOVA was performed to provide a crucial developmental context, specifically by identifying the age range at which significant performance improvements begin across different task conditions.

Comment 5: I suggest clear research questions, rationale behind each analysis pointing back to research questions, clear description of the answer to each RQ, and then unpacking in the results.  This would provide a much clearer understanding of the purpose of the study and the results.

Response 5: Thank you for this highly constructive suggestion. We fully agree that a clear and explicit link between our research questions and the analyses performed is essential for a well-structured manuscript. We have revised the Results section to directly address this point. The section is now structured to provide the rationale for each analysis, explicitly stating which research question it was designed to answer.

Comment 6: Some of the phrases felt a little odd or peculiar.  For example, we do not refer to an analysis by the name in the possessive (i.e. McNemar's analysis).

Response 6: Thank you for pointing out the issues with phrasing, specifically the use of the possessive 's' with analysis names. We have gone through the entire manuscript and corrected all instances of this error (e.g., "McNemar's analysis" is now "McNemar analysis").

Comment 7: Also unsure of the use of the word Kindergarten.  Ages 3-6 preschoolers cannot come from kindergarten.  Kindergarten is a class. 

Response 7: We appreciate you highlighting this important terminological ambiguity. As you correctly noted, the term "kindergarten" has a specific meaning in many educational systems that does not align with the 3-6 age range. We have therefore replaced the word "kindergarten" with the more appropriate term "preschools" throughout the manuscript and added a clarification of the age range to avoid any cross-cultural confusion.

Comment 8: I would rely more on a Native English speaker who understands content and context of the paper throughout rather than any machine translations, which it seems like machines were used.

Response 8: We appreciate your feedback regarding the clarity and natural flow of the English. We have addressed all of the specific phrases you highlighted. The manuscript has now been thoroughly reviewed and revised to ensure that the language is fluent, precise, and academically appropriate.

We believe that these revisions have addressed all of your concerns and have significantly strengthened the manuscript. Thank you once again for your constructive and insightful comments. We hope that the revised manuscript is now suitable for publication.

Sincerely,

Melis Süngü

Reviewer 3 Report

Comments and Suggestions for Authors

Dear Authors, 

your manuscript is dealing with a core topic in developmental psychology, that of children's social understanding and mentalizing capacities. Below you can find some suggestions that will help you to improve the quality of your work:

Introduction: You must ackowledge that FB reasoning requires basic representation based understanding of both the behaviour and the mind. It is mainly a mentalizing ability. Explain better the reason why you have chosen to focus only on the one of the three classic FB tasks, that of the 'unexpected transfer'. The other two (the 'deceptive box' and the 'deceptive object') which assess children's understading of FB in themselves are also widely used, and reliable tools in the investigation of social understanding development. The distinction between FB and PAR is not clearly presented. Do both measure the same ability? If not, why PAR is presented as an ulternative way of evaluating ToM in young children? The part of the Introduction which discusses the contribution of your study in the investigation of FB in children is confusing and not clarly stated. You major aim is to propose a more reliable way of assessing children's ToM or to highlight different aspects of children's ToM and propose a battery of tools that grasp them? The paper is missing your objectives and your resaerch hypotheses. In general, the Introduction needs reorganisation and a solid theoretical background which encompasses existent knowledge and highlights gaps and contradictions. It also needs to include a developmental percpetive of ToM and data on the role of language (since you examine different age groups and the vocabulary). Please include more recent references about the relation betawwen ToM and the vocabulary).

Methods: Remove from The Participants sections all information about Procedures and materiasl. The description of the tasks and conditions, although detailed, is confusing and not informative enough in terms of key differentiating features and objectives. It seems that the 4 conditions used in the study do not share any common ground, assess different outcomes and can not be compared.

Results: The main conditions FB, RR and PAK do not appear in the results. The reader is left with the impression that the above conditions (which are alternative according to the authors) will be compared and evaluated. 

Discussion: Discuss your results following the research hypotheses and include implications for child psychology and education.

Author Response

Dear Reviewer,

Thank you for your very detailed and insightful comments on our manuscript. We sincerely appreciate the time and effort you invested in providing such a thorough analysis of our work. Your suggestions have been invaluable in significantly improving the quality and clarity of our paper.

We have carefully addressed each of your points and have revised the manuscript accordingly. Below is a detailed, point-by-point response outlining the changes we have made.

Comment 1: Explain better the reason why you have chosen to focus only on the one of the three classic FB tasks, that of the 'unexpected transfer'. The other two (the 'deceptive box' and the 'deceptive object') which assess children's understanding of FB in themselves are also widely used, and reliable tools in the investigation of social understanding development.

Response 1: We appreciate the reviewer's request for clarification on our methodological choices. We have revised the Introduction to provide a more explicit justification for our decision to focus on the "unexpected transfer" paradigm. The revised text now explains that this particular version of the task was chosen because it is the most widely used and validated in the literature. More importantly, its structure allows for the precise manipulation of a target object’s location and presence, which is the central methodological innovation of our study.

Comment 2: The distinction between FB and PAR is not clearly presented. Do both measure the same ability? If not, why PAR is presented as an alternative way of evaluating ToM in young children?

Response 2: We appreciate the reviewer for highlighting this critical point. To address this, we have revised the relevant paragraph in the Introduction to more explicitly define and differentiate these two concepts. We now explain that while both strategies can lead to a correct answer on the FB test, they measure fundamentally different cognitive abilities. By providing a clear example, we illustrate how a child using PAR can pass the test without a true understanding of another's false belief.

Comment 3: The part of the Introduction which discusses the contribution of your study in the investigation of FB in children is confusing and not clearly stated. Your major aim is to propose a more reliable way of assessing children's ToM or to highlight different aspects of children's ToM and propose a battery of tools that grasp them?

Response 3: We agree with the reviewer that the original text did not sufficiently clarify the primary contribution and dual aims of our study. To address this, we have extensively revised the final paragraph of the Introduction. The updated text now explicitly states that our study's aim is twofold: to propose a more reliable assessment tool by systematically manipulating the perceptual properties of the false belief task, and to provide a more nuanced understanding of children's ToM skills by elucidating the distinct reasoning strategies (e.g., Belief Reasoning vs. Reality Reasoning) they employ.

Comment 4: The paper is missing your objectives and your research hypotheses.

Response 4: We agree that our initial introduction lacked a clear purpose and research hypotheses. We have now reorganized the Introduction to include a dedicated section at the end that clearly outlines the study's primary objectives and specific research hypotheses. This provides a solid framework for the rest of the paper.

Comment 5: In general, the Introduction needs reorganization and a solid theoretical background which encompasses existent knowledge and highlights gaps and contradictions.

Response 5: Thank you for this valuable feedback regarding our Introduction. We agree that a clearer structure and a stronger theoretical foundation were necessary. We have significantly revised the Introduction to directly address your concerns. The section is now reorganized to logically build a case for the study. We have also added a clear statement of our research questions and hypotheses, which explicitly highlights the gaps and contradictions that our study aims to address. We believe these changes provide a more cohesive and compelling foundation for the paper.

Comment 6: It also needs to include a developmental perspective of ToM and data on the role of language. Please include more recent references about the relation between ToM and the vocabulary.

Response 6: Thank you for your valuable feedback regarding the inclusion of a developmental perspective on Theory of Mind (ToM) and the role of language. We fully agree that these are critical topics, particularly when examining a sample of varying ages. Our manuscript's primary focus is on exploring the nature of false belief (FB) skills based on object presence and location, and proposing a more reliable assessment tool. A comprehensive investigation into the developmental trajectory of ToM and its relationship with language development was not the central aim of our study. We also received feedback from other reviewers encouraging us to condense our manuscript to focus more directly on our core findings, which led us to intentionally limit the scope of the developmental and language-related discussion. However, in line with your suggestion, we have revised the Introduction to more clearly articulate the developmental context and the role of language within the scope of our research. Specifically, we have included more recent references on the relation between ToM and vocabulary to strengthen the theoretical background. We believe these revisions provide a more complete picture of our findings while maintaining the paper's primary focus on the influence of object salience on FB reasoning.

Comment 7: Remove from The Participants sections all information about procedures and materials.

Response 7: We appreciate the reviewer's suggestion to enhance the Method section. We have restructured the whole section to ensure a clear separation between the "Participants" and "Materials and Procedures" sections.

Comment 8: The description of the tasks and conditions, although detailed, is confusing and not informative enough in terms of key differentiating features and objectives. It seems that the 4 conditions used in the study do not share any common ground, assess different outcomes and cannot be compared.

Response 8: We appreciate your valuable feedback regarding the clarity of our tasks and conditions. We agree that a more explicit explanation of their differentiating features and our rationale for their use was necessary. We have made significant revisions to address this point directly. The revised "False Belief Tasks" subsection now explicitly explains that all three tasks are systematic variations of the same well-established transfer paradigm. The Standard FB Test is framed as the baseline, and the three alternative tasks are presented as controlled manipulations of key perceptual features. Furthermore, we have ensured that the key distinguishing features of each condition are more clearly highlighted within the "Coding and Data Analysis" section to improve overall clarity and demonstrate their comparability.

Comment 9: The main conditions FB, RR and PAK do not appear in the results. The reader is left with the impression that the above conditions (which are alternative according to the authors) will be compared and evaluated.

Response 9: We thank the reviewer for this invaluable comment, which highlighted a critical point of ambiguity in our manuscript. We agree that the original Results section created confusion by not clearly distinguishing between the experimental tasks/conditions and the reasoning strategies we measured. To address this, we have revised the last paragraph of the Coding and Data Analysis section to establish a clearer logical flow. The updated text now explicitly states that the four distinct false belief tasks were designed to elucidate the specific reasoning strategies (Belief Reasoning, Perceptual Access Reasoning, and Reality Reasoning) children employ. This revision clarifies that we are not comparing a set of disparate conditions, but rather analyzing how our systematic manipulations of the tasks influenced the use of these underlying cognitive strategies. We believe this change now guides the reader more effectively through our findings and fulfills the promise of our research design.

Comment 10: Discuss your results following the research hypotheses and include implications for child psychology and education.

Response 10: We appreciate the reviewer's valuable feedback. We agree that our Discussion section could be more effective by being more directly aligned with our research hypotheses and by including practical implications. We have made significant revisions to address these points. The Discussion section has been restructured to explicitly address each of our research hypotheses in separate, dedicated paragraphs. The revised text now clearly states whether our findings provide support for each hypothesis, directly linking our results to our core arguments. Also, we revised the final paragraph of the Discussion to clarify on the implications of our findings for child psychology and education. This section highlights how our results can inform the development of more nuanced assessment tools and guide educational practices, moving beyond a simple pass/fail evaluation of children's abilities.

We are confident that these comprehensive revisions have addressed all of your concerns and have significantly improved the overall quality of our manuscript. We sincerely thank you for your thoughtful and constructive feedback.

Sincerely,

Melis Süngü

Round 2

Reviewer 2 Report

Comments and Suggestions for Authors

This is much better- the analyses have improved very much so.  My only comment and I defer to editors is that the references are very old, there seems to be an addition of a few ~3-4 new references ranging from 2021-2023, but the majority of the references are quite old and the editors should decide if this is something they wish the authors to improve on.   I am satisfied with the improvements to the methods, which is my area of expertise. 

Author Response

Comments 1: This is much better- the analyses have improved very much so.  My only comment and I defer to editors is that the references are very old, there seems to be an addition of a few ~3-4 new references ranging from 2021-2023, but the majority of the references are quite old and the editors should decide if this is something they wish the authors to improve on.   I am satisfied with the improvements to the methods, which is my area of expertise. 

Response 1: Dear Reviewer,

Thank you for your valuable time and positive feedback on our manuscript. We are delighted that you are satisfied with the improvements made to our analyses and methods. We have carefully considered your comment regarding the age of our references. We wish to clarify that the foundational works in our introduction were deliberately included to provide essential historical context and theoretical grounding for our study. Furthermore, the new references ranging from 2015 to 2017 were added to engage with the current methodological debate in the field and directly address the points raised in the previous review. We believe that our reference list now offers a balanced approach, encompassing both the classic literature that established the field and the more recent sources that reflect its ongoing evolution. Nevertheless, we are of course willing to update our references further if the editor deems it necessary to improve the quality of the manuscript.

Thank you once again for your constructive and insightful comments.

Melis Süngü

Reviewer 3 Report

Comments and Suggestions for Authors

Dear Authors, the revised version of your paper is much improved. However, there are still issues which need to be addressed carefuly so that several methodological and reliability concerned are eliminated.

  1. The argument that you chose to focus on the 'unexpected transfer' task due to its high reliability as compared to the other two classic false belief tasks, is weak. It is weak because all the 3 tasks are widely used in the literature for many years now, and, also because they measure different aspects of a child's social understanding capacity. It is very rarely that we meet in the literature studies using only one task.
  2. It is still unclear and confusing how the classic FB condition differ from the Basket Condition (Reality & Belief, Diverge).
  3. You need to address developmental issues, since you present your data adopting a developmental perspective. 

Author Response

Comments 1: The argument that you chose to focus on the 'unexpected transfer' task due to its high reliability as compared to the other two classic false belief tasks, is weak. It is weak because all the 3 tasks are widely used in the literature for many years now, and, also because they measure different aspects of a child's social understanding capacity. It is very rarely that we meet in the literature studies using only one task.

Response 1: Thank you for your valuable and insightful feedback on our manuscript. We agree that different false-belief tasks measure distinct aspects of children’s social understanding. The primary aim of our study was not to provide a general assessment of Theory of Mind, but rather to isolate and examine a key confounding factor: the perceptual salience of the object. We chose this paradigm because it is uniquely suited for this investigation, as it inherently contains the very confounding variable we sought to manipulate. We believe this focused approach allows our findings to make a direct and clear contribution to the ongoing debate about the validity and limitations of classic false-belief tests. To further clarify this methodological choice, we have added a sentence in the introduction of our revised manuscript to explicitly state our research objective.

Comments 2: It is still unclear and confusing how the classic FB condition differ from the Basket Condition (Reality & Belief, Diverge).

Response 2: Thank you for your constructive feedback. Your comment has helped us identify a crucial point that needed to be more explicitly stated in the manuscript. We agree that while both tasks involve a divergence between a character's belief and reality, the fundamental difference lies in the reason for the false belief. In the Standard FB Test, the doll's false belief is based on a lack of information—she is unaware of the ball's transfer. In the Basket Condition, the doll's false belief is based on a failed expectation—she believes the ball has been moved as she requested. To address this, we have revised our manuscript to clarify this conceptual distinction. We have added a new sentence at the “2) Basket Condition (Reality & Belief Diverge)” section of the Methods, explaining that our tasks were designed to differentiate between these two distinct types of false belief. Furthermore, we have revised the corresponding paragraph in the Discussion to highlight that the differing results between the two conditions underscore the context-dependent nature of children’s reasoning.

Comments 3: You need to address developmental issues, since you present your data adopting a developmental perspective. 

Response 3: Thank you for your valuable feedback. We have carefully considered your point that our manuscript needs to more explicitly address developmental issues, especially given the inclusion of age and language data. To address this, we have revised the Discussion section to more thoroughly integrate our developmental findings with our primary cognitive and methodological arguments. The new text clarifies how the observed age-related performance differences—particularly the disparity between the standard and alternative tasks—serve as a crucial validation of our claim that misleading perceptual cues and other task demands hinder children's true ToM capacity. Furthermore, we have reframed the discussion on the temporal lag between receptive vocabulary and FB understanding. We believe this revision provides a more cohesive narrative, positioning our developmental data not as a separate analysis, but as integral evidence supporting our core contribution.

Thank you once again for your insightful comments.

Melis Süngü

Round 3

Reviewer 3 Report

Comments and Suggestions for Authors

Dear Authors 

Your manuscript is much improved. You have responded adequately to most of my comments. My only concern remains with your decision not to include in your method all the 3 classic false belief task since all of them involve the examined parameter of the perceptual salience of the object. I suggest that you acknowledge thia as a key limitation of the study. 

Author Response

Comment 1: Dear Authors 

Your manuscript is much improved. You have responded adequately to most of my comments. My only concern remains with your decision not to include in your method all the 3 classic false belief task since all of them involve the examined parameter of the perceptual salience of the object. I suggest that you acknowledge thia as a key limitation of the study. 

Response 1: Dear Reviewer,

Thank you for your valuable final comment. We agree that our manuscript is much improved, and we appreciate your careful reading and feedback. We understand your concern regarding our decision to focus on the unexpected transfer paradigm. We agree that this represents a key limitation of the study. To address your suggestion and enhance the rigor of our paper, we have added a new paragraph in the Discussion section to explicitly acknowledge this limitation.

Thank you once again for your constructive guidance throughout this review process.

Sincerely,

Melis Süngü